# An automated platform for high-throughput mouse behavior and physiology with voluntary head-fixation

Ryo Aoki[1], Tadashi Tsubota[1], Yuki Goya[1] & Andrea Benucci[1]

Recording neural activity during animal behavior is a cornerstone of modern brain research. However, integration of cutting-edge technologies for neural circuit analysis with complex behavioral measurements poses a severe experimental bottleneck for researchers. Critical problems include a lack of standardization for psychometric and neurometric integration, and lack of tools that can generate large, sharable data sets for the research community in a time and cost effective way. Here, we introduce a novel mouse behavioral learning platform featuring voluntary head fixation and automated high-throughput data collection for integrating complex behavioral assays with virtually any physiological device. We provide experimental validation by demonstrating behavioral training of mice in visual discrimination and auditory detection tasks. To examine facile integration with physiology systems, we coupled the platform to a two-photon microscope for imaging of cortical networks at single-cell resolution. Our behavioral learning and recording platform is a prototype for the next generation of mouse cognitive studies.

[1] RIKEN Brain Science Institute, Wako-shi, Saitama, 351-0198, Japan. Ryo Aoki and Tadashi Tsubota contributed equally to this work. Correspondence and requests for materials should be addressed to A.B. (email: andrea.benucci@riken.jp)

Understanding the complexity of biological systems is a profound challenge for science and technology. In neuroscience, elucidating the principles governing the production of complex behavior in mammals from the vast tangle of connected neural circuits is one of the most difficult problems. Conventional methods such as those using head-fixed assays require tedious and labor-intensive brain recordings from single animals engaged in behavioral tasks over weeks to months. However, the specificities of these paradigms and their integration with the growing array of state-of-the-art brain physiological recording systems differ greatly among and within laboratories due to the variability introduced by the experimenter's intervention. This lack of standardization generates inherent reproducibility issues and eliminates the possibility of large, sharable data sets that could significantly accelerate the pace of scientific discovery and validation. These problems have recently become apparent in mouse studies. Among mammals, the mouse contains the largest methodological toolbox for neural circuit research on behavior. Accordingly, researchers are training mice in complex behavioral assays with concurrent physiological recording and manipulation (e.g., multi-channel electrophysiology, imaging, optogenetics[1, 2]). However, training mice to learn complex behavioral tasks requires time-consuming species-specific training methods that stem from innate phenotypic and behavioral characteristics. Indeed, even within rodents, mice have unique characteristics including high sensitivity to experimenter biases[3] and physiological stresses from handling[4]. Several mouse behavioral systems have been reported that attempt to address experimental limitations associated with the short mouse life cycle and the decreasing yield of trained animals with the increasing complexity of the behavioral tasks[5–11].

Among the proposed solutions, some rely on the experimenter's intervention, e.g., for head fixation[6, 7, 9, 11], while others use freely moving animals[5, 10, 12], with a noteworthy recent report

of a setup featuring self-head fixation for automated wide-field optical imaging[8]. From a research economics perspective, an ideal mouse system would feature self-head fixation for behavioral training and rapid exploration of a large space of complex behavioral parameters with minimal experimenter intervention, allow high-throughput automated training, have the capability to explore various sources of psychometric data, flexibly integrate multiple physiology recording/stimulation systems, and enable the efficient generation of large, sharable, and reproducible data sets to standardize procedures within, and across laboratories.

We developed an experimental platform for mouse behavioral training, with full automation, voluntary head fixation, and high-throughput capacity. The platform is scalable and modular allowing behavioral training based on diverse sensory modalities, and it readily integrates with virtually any physiology setup for neural circuit- and cellular-level analysis. Moreover, its remote accessibility and web-based design make it ideal for large-scale implementation. To demonstrate the optimality of the system for the integration of complex behavioral assays with physiology setups, we used the platform to train mice in two behavioral tasks, one visual and one auditory. The training was compatible with stable cellular-level imaging as we demonstrated with two-photon GCaMP recordings in trained animals.

## Results

**Habituation system for head restraining.** In both behavioral tasks, newborn mice were initially housed in an enriched environment. At the age of ~P45 they were implanted with a head-post and a round chamber, and individually housed in standard cages (Supplementary Table 1). After recovery, mice were put on a water-controlled regime (~1 mL/day) for about 1 week. Then a self-head-restraining device was introduced to the home cage (Fig. 1a; Supplementary Fig. 1) that is fully compatible with

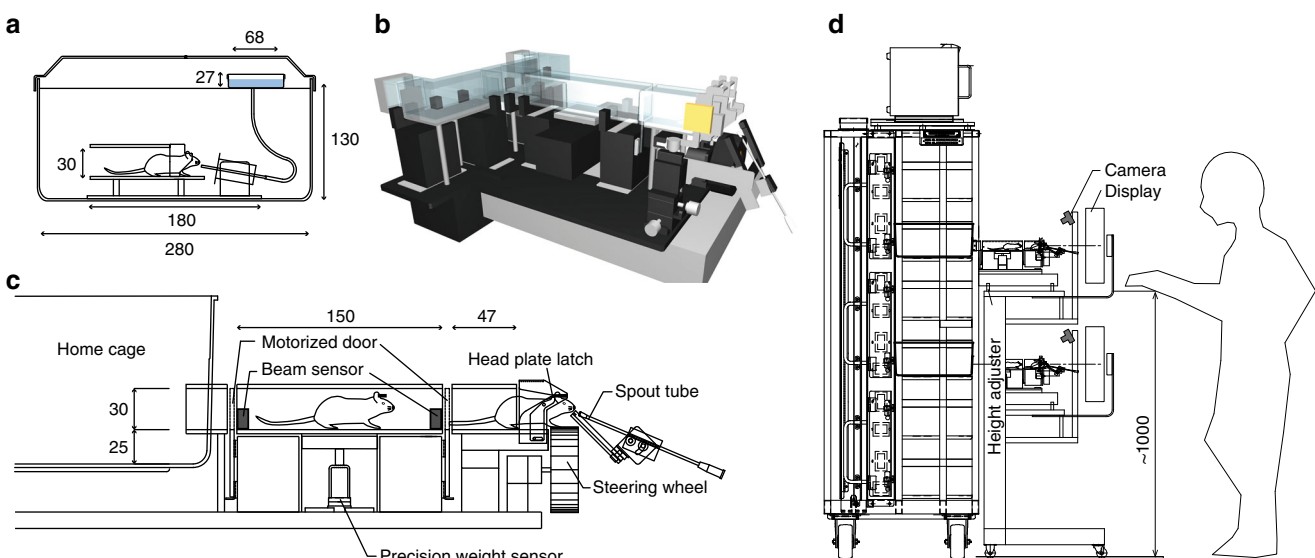

**Fig. 1** High-throughput automated setup for voluntary head fixation. **a** Head-restraining habituation system. Depiction of a mouse inside a transparent PVC tube drinking water from a water spout connected to a water tank (blue box). Narrowing rails on the sides of the tube progressively restrain the head-post (details in Supplementary Fig. 1). There is no latching mechanism for the head-post, hence mice can back out of the tube at any time. The containing box represents a mouse cage with an air-filtering lid and a metal grid for mouse containment (horizontal solid line under the lid) as in standard individually ventilated cages. Over the course of a couple of days, mice routinely self-restrained to drink water. **b** 3D view of the main dual cage setup (details of all hardware and software components in the Supplementary Material). **c** Side view of setup and labeling of its main components. **d** Side view of two setups (training capability of a single setup is four mice/day) housed in a mouse rack. Several setups can be accommodated in the same rack (current capacity: 12 platforms, 48 mice/day, ~12,000 trials/day). All units are in mm. Original technical drawings edited with permission from O' Hara & Co., Ltd.

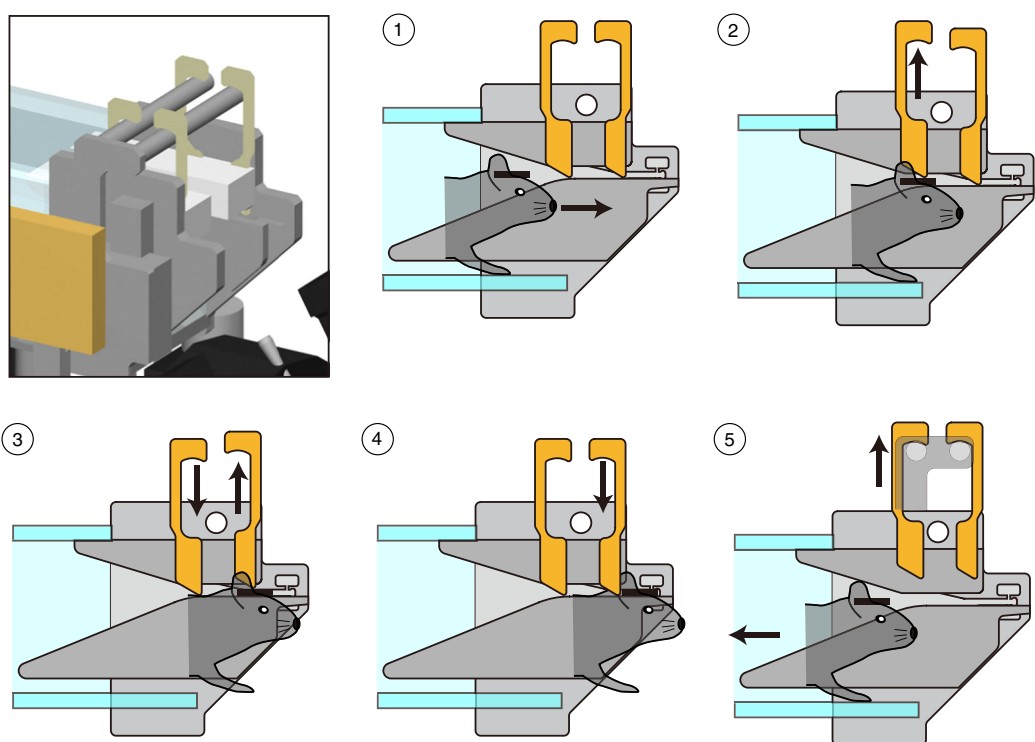

**Fig. 2** Automated self-latching. Top-left panel shows a 3D rendering of the latching mechanism (details of the components and how they are assembled are provided in Supplementary Fig. 3). Panels 1–5 (circled numbers) show the sequence of steps leading to self-head fixation: (1) the head-plate (black bar on mouse head, Supplementary Fig. 1) is progressively restrained by narrowing rails (gray converging lines, Supplementary Fig. 3). (2) The forward motion of the head-plate mechanically lifts up the first pair of latching pins. (3, 4) The first pair of pins then lowers by gravity, and the continued forward motion of the animal similarly lifts up and down the second pair of latching pins, leading to the final self-head fixation (4). During 3, 4, small tilt and forward movements are allowed that reduce the probability of a "panic" response due to a sudden head fixation. (5) When the task session ends, a computer-controlled servo motor actuator lifts up both pairs of latching pins and releases the animal (Supplementary Fig. 4). Original technical drawings edited with permission from O' Hara & Co., Ltd.

standard mouse cages and IVC racks (Fig. 1d). Mice learned that to obtain water they had to restrain their head-post, but without latching it (Supplementary Fig. 1). This step is key for future self-head fixation; if head-fixed without habituation, mice form a strong initial aversive association and then avoid the setup in following sessions, as typically observed in fear conditioning paradigms. During this habituation phase, the body weight of animals was monitored twice/day and mice were removed from training if their weight dropped below 70–75% of their original body weight (corrected by gender and age).

**Main setup**. After 1 week, the habituation system for head restraining was removed and the cage was replaced with a similar one having a small (4 × 4 cm) aperture in the front to give the animal access to the main training setup (Fig. 1b, c; Supplementary Fig. 2). This behavioral training setup for voluntary head fixation consists of (1) hardware components connecting the home cages to the latching unit, (2) a circuit board interface to activate several electronic components, and (3) two software controllers (Supplementary Figs. 2–5; Supplementary Tables 2, 3). The (passive) hardware components consist of a T-shaped assembly of transparent PVC tubes connecting two mouse cages (housed in a standard IVC mouse rack) to a latching unit (Fig. 1b, c; Supplementary Fig. 2). Access to the tubes is controlled by motorized duralumin doors that prevent mice from entering other cages. Several electronic components are positioned along the tubes: IR sensors allow precise monitoring on the animal position when in the setup. A scale positioned in the

middle of the central tube is used for automated weight measurements. At the end of the central tube, a servo-controlled latching system is used to fix the head-post. The latching system was designed to minimize the animal's stress due to head fixation (Fig. 2; Supplementary Figs. 2, 3). As the animal advances to reach the water spout, the forward movement of the leading edge of the head-post mechanically lifts two small metal pins positioned to the left and right side of the tube, on top of the restraining rails (Fig. 2). The pins lower by gravity right behind the trailing edge of the head-post, thus stopping the animal from moving backward. At this stage, the head is not fully restrained: the animal can move (forward) and can make small head-tilt movements. These degrees of freedom are small enough to prohibit any escape back to the home cage, but sufficiently large to avoid a panic reaction, typically accompanied by loud vocalizations and jerky body movements. Eventually, mice advance and encounter a second set of pins that will stably latch the head-post at the very end of the rails, positioning the mouse in front of the water spout. A latching sensor informs the central software to initiate the training session. Informed via TCP/IP communication, a second software (Matlab, Psychophysics Toolbox v3.0.12, The MathWorks, Inc., Natick, MA, see also Supplementary Table 3) takes over, controlling the input devices, the behavioral readouts, and the reward system. At the end of the training session, a servo motor actuator lifts both sets of latching pins, allowing the animal to return to the home cage (Supplementary Fig. 4; Supplementary Movie 1). Seven infra-red beams distributed in key locations inform the central software of the animal's position, making sure the mouse is never trapped in

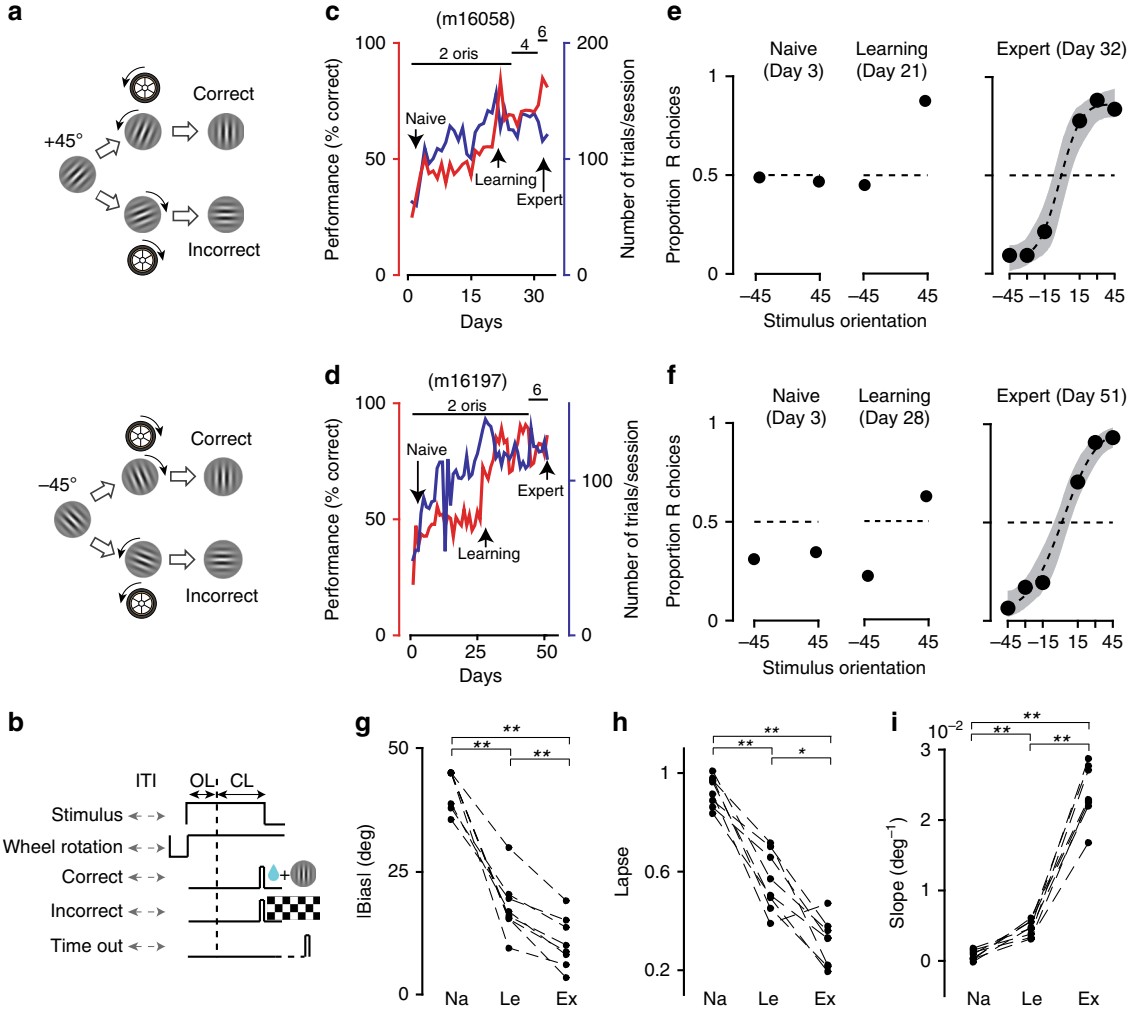

**Fig. 3** Two-alternative forced choice visual discrimination task. **a** Example trial: a sinusoidal static grating with +45° clockwise (top) or −45° counterclockwise (cc, bottom) orientation from vertical was presented on the screen. Mice had to indicate whether the orientation was clockwise or cc by rotating with the front paws a wheel position between them and the screen, with the wheel rotation controlling the real-time close-looped orientation of the visual stimulus. **b** Trial structure of reward and negative feedback; ITI inter trial interval (randomized), OL open-loop period, CL close-loop period. **c** Changes in performance and number of trials over training days for an example mouse (m16058), colors reflect the left–right y-axes. Top horizontal bars and numbers indicate changes in the number of orientations the animal had to discriminate: two orientations (±45, **a**), up to six orientations for a minimum deviation from vertical of ±15°. The number of orientations increased when performance reached ~70%. **d** Same as **c** for a different example mouse (m16197). **e**, **f** Psychometric curves during three different learning phases, naive, initial learning, expert, for m16058 (**e**) and m16197 (**f**). Shaded area in the expert curves represent 95% bootstrapped confidence interval. **g–i** Changes in bias, lapsing rate, and slope derived from psychometric functions (and from a linear model for two orientation conditions, Methods) during learning (n = 8 mice). Each dot is an average across the first 6 days of the corresponding learning phase (Na, Le, and Ex for naive, learning, and expert). P-values are calculated from two-tailed Wilcoxon signed rank tests, **P < 0.01, *P < 0.05

between motorized doors (Supplementary Table 3). In front of the latching unit are a water spout, a low-friction rubber wheel (Lego, $D = 62.4$ mm, $W = 20$ mm) attached to a rotary encoder, and an LCD display (LG, refresh rate: 60 Hz, ~25 cm distant from latching mechanism and gamma corrected using a photometer to linearize the luminance of the color channels) (Supplementary Table 2). For the auditory task, in addition, a speaker is placed in front of the latching unit. Custom-made software controls the hardware (motorized doors, scale, IR sensors, head-latching sensor, head-releasing actuator) via a DAQ board (NI USB-6501, National Instruments), thus regulating access to the setup. An automated e-mail system alerts the experimenter if the animal's weight drops below a pre-determined threshold. The second software package runs on a laptop and controls the task session. It is triggered by the first software after the animal has self-latched, where it (1) controls the visual stimuli presented on the LCD

monitor or the auditory stimuli, (2) reads the output of rotary encoder, and (3) triggers a pump for water delivery. A USB webcam (C525, Logitech) is used to monitor the animal's behavior during each session and can be accessed using a remote video connection (Skype, Microsoft). Overall, mouse access to the setup, the latching, and unlatching mechanisms are all controlled by a computer, and thus is fully automated. A training session lasted for a fixed duration of 20–30 min and consisted of $119 \pm 7$ trials for the visual task (s.d., n = 8 mice that learned the tasks out of 12 entering the pre-training phase), and $119 \pm 20$ trials for the auditory task (n = 2 mice that learned the tasks out of 2 entering the pre-training phase). Mice were given access to the setup twice a day, with a 4.5–6 h interval. A technician replaced mouse cages at the beginning and at end of the day, thus allowing two mice/ setup to be trained during the day and two mice/setup during the night. Hence a single setup can currently train four mice in 24 h.

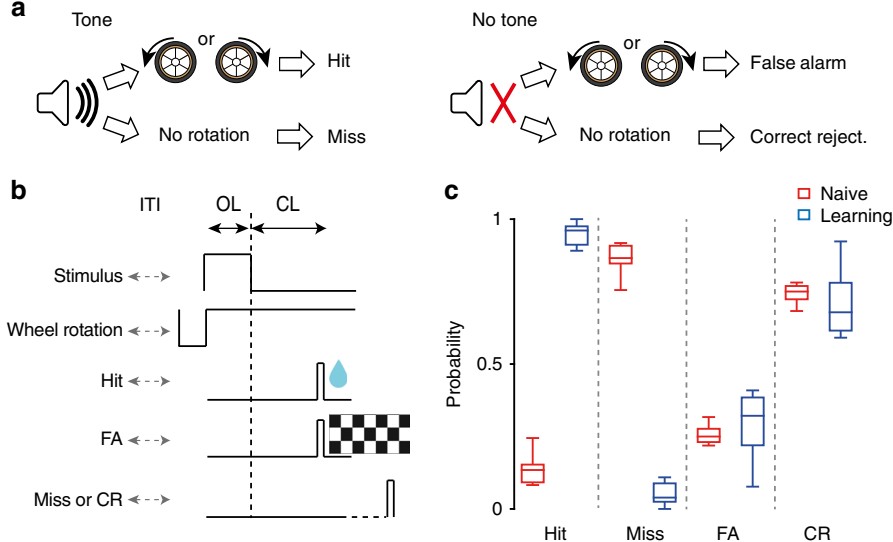

**Fig. 4** Go-no-go auditory detection task. **a** Example trial: in 70% of the trials, the animal was presented with brief tones (left panel). Mice had to respond by rotating the wheel in either direction. In the remaining 30% of trials, no tone was presented and the mouse had to refrain from rotating the wheel (right panel). **b** Trial structure of reward and negative feedback; ITI inter trial interval (randomized), OL open-loop period, CL close-loop period. **c** Box-and-whisker plot comparing percentage of Hit, Miss, false alarm (FA), and correct rejection (CR) responses of a representative mouse for first 10 sessions (Naive) and 15 consecutive sessions after the mouse reached a performance of $d' > 1.5$ (learning). The whiskers show the minimum and maximum of data distribution; box lines show the 25th percentile, the median, and the 75th percentile. In naive mice, FA rates are higher than Hit rates possibly due to a startle reflex following the target sound presentation ($P < 0.005$, $n = 10$ sessions, Wilcoxon signed rank test)

Two sessions per day (or three for the orientation discrimination task described below) thus produce ~1000 trials/setup/day. Several setups can be accommodated in a standard mouse rack to achieve high-throughput capacity (Fig. 1d). At a current capacity of 12 platforms, we can train 48 mice/day and collect up to ~12,000 trials/day. This number could easily be increased if the cages were replaced more frequently. The hardware and software design has proved to be safe for the animals with no reported accidents with over ~100 mice trained in the setup to date.

**Behavioral training 1: orientation discrimination**. We demonstrated the ability of our setup to train mice in two behavioral tasks based on different sensory modalities: vision and audition. For vision, we trained mice to perform in a two-alternative forced choice orientation discrimination task relying on binocular vision[13]. We designed a 2D interactive visual task in which a circular grating placed in the central part of the visual field had a clockwise (c) or counterclockwise (cc) rotation relative to vertical (Fig. 3a). The grating diameter was small enough (20° diameter, $n = 1$ or 30°, $n = 11$) to engage only the V1 binocular region. This task did not rely on first-order visual features, e.g., luminance or contrast that were kept constant, but on a second-order feature, the grating orientation (Methods). For trial initiation, mice had to keep their front paws on a small wheel[6] and refrain from making wheel rotations for 1 s (within ±15°). A stimulus was then shown on the screen statically for 1 s during which time possible wheel rotations were ignored by the software (open loop) (Fig. 3b). After this open-loop period, mice reported their percept of the stimulus orientation with c/cc rotations of the wheel for corresponding c/cc rotations of the grating stimulus, with the wheel rotation controlling the orientation of the visual stimulus in real time (closed loop). A correct response was a c (or cc) rotation to a cc (or c) rotated stimulus, resulting in a vertically oriented grating, the target orientation. After a correct response, the vertically oriented grating remained on the screen for an additional 1 s to promote the association between the vertical orientation and the reward. Correct responses were rewarded with a small amount of

water (typically 4 µL, but sometimes adjusted within a range of 2–4 µL according to the body weight of the animal). Incorrect responses were punished with a 5 s time-out stimulus consisting of a full-field, flickering square-wave checkerboard with 100% contrast. If there was no rotation crossing a near-vertical threshold of 10° for 10 s after the onset of the closed loop, the visual stimulus disappeared and the next trial started (time-out trials, Fig. 3b). The inter trial intervals (ITI) had a random duration of 2–5 s. During the initial phase of the training, a grating stimulus with +45° or −45° orientation was presented in each trial. Once a mouse reached ~70% accuracy, the task difficulty was increased by adding new orientations (−30°, −15°, +15°, +30°) (Fig. 3c, d). An adaptive corrective procedure was adopted during training to help mitigate rotation biases (Methods). Independently, we also set a limit for session duration (~20 min). Mice performed three sessions per day and learned the task, discriminating orientations as small as 15° from vertical. Eight out of twelve mice that entered the pre-training phase learned this task. Initial learning for the 45°/−45° orientation discrimination task took ~4 weeks (26 ± 7 days, s.d., $n = 8$), while it took on average ~8 weeks (59 ± 27 days) to reach a 75% accuracy with the smallest discrimination angle (±15°). We quantified the animals' performance with standard psychometric curves[14]. During the naive period, as expected, mice performed almost randomly, showing performance no better than chance level because of biases and with a large lapsing rate (Fig. 3e, f). As training progressed, all of the three psychometric parameters, bias, lapsing rate, and slope showed significant changes, synergistically reflecting the learning of the task (Fig. 3g–i). Accordingly, mice eventually reached a >75% performance level with reduced bias and lapsing rate, and with increased discrimination sensitivity (Fig. 3g–i).

**Behavioral training 2: tone detection**. To illustrate the use of our setup in tasks relying on different sensory modalities, we trained a group of mice in an auditory go-no-go task. We placed a speaker for auditory stimulation in front of the animal, ~10 cm from the

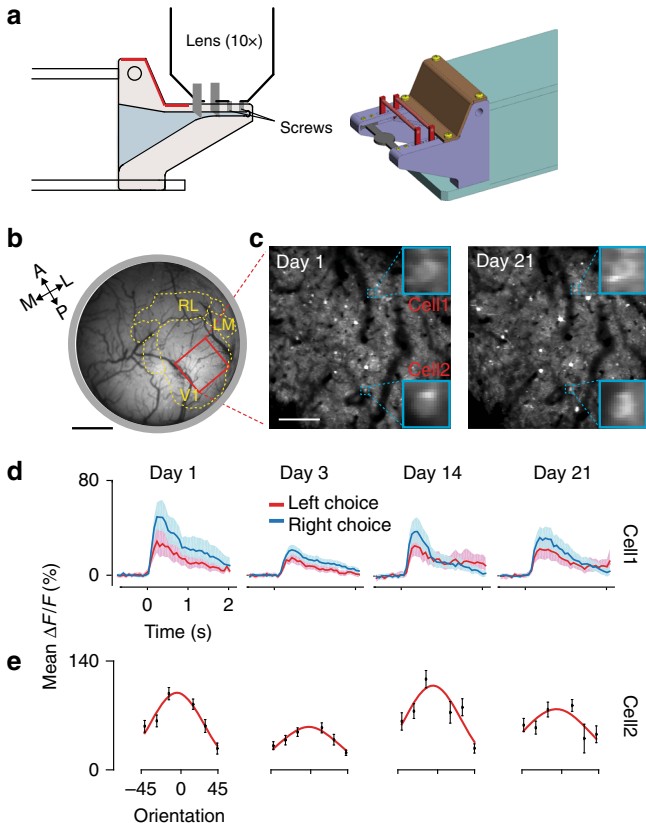

**Fig. 5** Compatibility of the setup with two-photon imaging. **a** Left panel: side view schematic of the latching unit for physiology with a ×10 objective lens for two-photon imaging. The two vertical gray segments indicate the latching pins (Supplementary Fig. 6). Right panel: 3D rendering of the latching part of the unit. **b** Wide-field GCaMP8 fluorescence image at the cortical surface. Red rectangle, ROI within V1. Scale bar 1 mm, dotted yellow lines and labels are segmented visual areas (Methods). **c** Repeated imaging of the same ROI across different days. Blue squares are magnified views of two example cells. Scale bar, 200 µm. **d** Visually evoked responses during the behavioral task recorded from cell1 shown in **c**. $\Delta F/F$ trial averages for left/right choices (red/blue lines). Shaded areas, 95% confidence interval. Time-out trials were not included in the data. **e** Responses to oriented gratings (30° diameter) from cell-2 shown in **c**. Error bars are s.e.m. Red curves, Gaussian fit to the data. Original technical drawings in **a** edited with permission from O' Hara & Co., Ltd.

mouse ears, and enclosed the setup in a sound isolation box to reduce ambient noise (Fig. 4; Methods). The transparent PVC tube connecting the cage to the main setup passed through an aperture on the side of the isolation box. Mice had to detect the occurrence of an 80 dB, 10 KHz pure tone played five times (for 500 ms, five pulses of 5 ms duration at 10 Hz[15]). This go stimulus was presented in 70% of the trials. In the remaining 30% of the trials, the mouse was exposed to an unmodulated ~50 dB background noise (no-go stimulus)[16]. Mice had a 2 s window from the end of the go-stimulus to report the tone detection by rotating a small wheel at least 70° in either direction (Fig. 4a). Typically, each mouse had a preference for clockwise or counterclockwise rotations. In hit trials (Hit, go responses to go stimuli) mice were water rewarded, and the following trial would start after an ITI of randomized duration, 5–10 s. In miss responses (Miss, no-go responses to go stimuli) mice did not receive any reward or punishment and a new trial would start after a randomized ITI duration. Similarly, in correct rejection trials (CR, no-go responses to no-go stimuli) mice were not rewarded nor

punished and a new trial would start after an ITI. In false alarm trials (FA, go responses to no-go stimuli), mice were punished by additional 10 s of ITI and shown a full screen, square-wave checkerboard with 100% contrast (Fig. 4b). Mice performed two sessions per day and learned the task, ($d' = 1.5$; Fig. 4c), over $12.5 \pm 3.5$ days (s.d., $n = 2$). FA rates remained constant throughout training, with FA rates in naive mice higher than hit rates ($P < 0.005$, Wilcoxon signed rank test) possibly due to a startle reflex following the target sound presentation[17–20].

In summary, these two tasks show that our setup can train mice in sensory-based decision making tasks, across sensory modalities, for accurate psychophysical measurements and with full automation.

**Latching unit for physiology.** In order to serve as a powerful tool to study the neural basis of cognitive functions, a setup must not only enable behavioral training but also (1) be easily integrated with diverse customized physiology setups, and (2) preserve the animals' behavioral performance in spite of integration specificities. To address the first point, we opted for a semi-automated solution, reasoning that full automation of physiology experiments is often unnecessary and sometimes even undesirable or unfeasible (e.g., when patching small cellular processes, repeatedly inserting fragile/bendable optic fibers, electrodes, silicon probes, etc.). Moreover, the bottleneck in a typical behavioral study is often the time required to train mice, while statistical power for the physiological experiments can typically be achieved with a handful of trained animals. Hence, we developed a semi-automated method relying on a movable unit (by the experimenter), which still had an automated self-latching mechanism and preserved the separation between the animal and the experimenter (Fig. 5a; Supplementary Fig. 6). Regarding the second point, we showed that by using such a unit, and by simply having input and output devices matching those used in the main training setup, mice readily adapted to the new environment (novel sounds, lights, and smells). As proof-of-principle, we demonstrated the integration of our setup with a two-photon microscope requiring cellular-level stability as the animal performed in the behavioral task.

**Two-photon imaging.** We trained mice expressing GCaMP8[21] (Methods) in the orientation discrimination task described in Fig. 3. When mice reached a threshold performance level (75% correct discrimination), we connected the latching unit for physiology to the home cage. Except for the reduced length of the main corridor, the unit was almost identical to the one used during training (Supplementary Fig. 6), and mice self-latched as before. At the end of the latching stage a set of four screws, tighten by the experimenter, allowed for stable block of the head-plate (Supplementary Figs. 6, 7). The platform was then placed under the two-photon microscope, still preserving the separation between the animal and the experimenter (Fig. 5a). Input and output devices in the two-photon setup matched those used in the training setup. However, the loud sounds generated by the galvo and resonant scanners, together with the bulky equipment on top of the mouse head were all novel elements. It took mice an average of $2.5 \pm 1.5$ sessions to return to the same performance level learned in the training setup ($n = 2$ mice, four sessions, and one session, respectively). Before commencing training, mice were imaged using standard methods for retinotopic mapping to identify V1 and higher visual areas[22] (Fig. 5b; Methods). In typical two-photon imaging experiments, we recorded from a volume $850 \times 850 \times 3$ µm³ of L2/3 neurons in the primary visual cortex (Fig. 5c). Using a common analysis for cell segmentation (Methods), we could identify ~200 neurons per volume. Using

vascularization landmarks, we could image the same cells over days or weeks (Supplementary Fig. 7), and segregated their responses as a function of the animal's choices (Fig. 5d) or stimulus orientations (Fig. 5e). As a corollary of this cellular-level resolution, our semi-automated procedure can then be easily combined with a large variety of other imaging, optogenetic, and electrophysiology systems requiring a similar degree of stability of the neural target of interest. In summary, the training setup combined with the latching unit for physiology is a convenient compromise for the relatively effortless integration of automated behavioral training with a large diversity of physiology systems.

## Discussion

The traditional experimental approach to integration of an animals' behavioral training and physiological recording has often resorted to lab-specific experimental configurations relying on the experimenter's intervention with the drawback of hindering within- and across-lab reproducibility. Here, in an effort to overcome these limitations, we describe an experimental platform for mouse behavioral training with high-throughput automation and voluntary head fixation that can integrate with diverse physiology systems, as we show with a two-photon microscope that requires cellular-level stability.

In developing this platform, we wanted to include three key features for behavioral training and imaging/electrophysiology systems: (1) stable head fixation, an important feature for behavioral assays relying on accurate measurement of many sensory modalities, such as vision that requires eye tracking and view point stability. Head fixation is also desirable for optical/optogenetic technologies aiming to achieve cellular-level resolution[1, 2]. (2) Full automation of the behavioral training, to remove the experimenter's subjective component along with the disadvantages of costly human labor and time required to collect sufficient data for reproducible science. (3) The ability to use trained mice within a broad spectrum of physiology systems for the interrogation of neural circuits. Our mouse platform simultaneously fulfills of all the above requirements, thus significantly facilitating the integration of a rich repertoire of cognitive paradigms with cutting-edge technologies for the detailed interrogation of neural circuits. The additional cost and time effectiveness of the platform is a key point in view of the need to quickly explore a large space of behavioral parameters for complex behavioral paradigms, a necessary but essential task that would be economically prohibitive without high-throughput automation. To demonstrate these experimental advantages, we trained mice in two behavioral tasks with voluntary head fixation. In the first task, we showed the automated training of mice in a 2AFC orientation discrimination task. In the second task, demonstrating the flexible use of the setup across sensory modalities, we trained mice in an auditory detection task. We finally demonstrated the ease of two-photon imaging of trained mice with the aid of a semi-automated latching unit.

Training mice in these tasks provides important validation benchmarks for the platform's performance. Considering the visual task, the learning yield was 67% (8 out of 12 mice) and the average training duration for peak discrimination performance was $59 \pm 27$ days (s.d., $n = 8$ mice). The task requires latching of the head-plate, daily monitoring of body weight, and incremental training procedures. If these steps were performed by a researcher, training in parallel four mice/day (a feasible commitment for a single person) the "human cost" for one trained mouse would be ~15 h (i.e., 22 days, with two 20 min sessions/day). In contrast, assuming all our setups were used for this task at our current capacity of 48 animals/day, we could produce an average of one trained mouse every 2 days (i.e., 1.3 h of rig time).

Even assuming that the presence of an experimenter could potentially accelerate the training thanks to a more rapid optimization of training parameters, it would still be difficult to achieve a 10-fold reduction in training duration as obtained with our setups. Furthermore, this reduction was achieved compatibly with two key requirements for this task: (1) head fixation for view angle stability across trials and eye tracking, and (2) control of the frequency and duration of the trials in those behavioral and physiology paradigms demanding session durations set by the experimenter rather than by the animal. In conclusion, we believe that our platform has many advantages to serve as an ideal system for the large-scale standardization of behavioral assays for facile integration with physiology systems.

## Methods

**Subjects**. All procedures were reviewed and approved by the Animal Care and Use Committees of the RIKEN Brain Science Institute. Behavioral data for the visual task were collected from eight C57BL/6J male mice, and from two Tg mice (Thy1-GCaMP6f (GP5.5)) for the auditory task. The age of the animals typically ranged from 8 to 28 week old from beginning to end of the experiments. Mice were housed under 12–12 h light–dark cycle. No statistical methods were used to predetermine the total number of animals needed for this study. The experiments were not randomized. The investigators were not blinded to the animals' allocation during the experiments and assessment of the outcome.

**Animal preparation for two-photon imaging**. Implantation of a head-post and optical chamber. Animals were anesthetized with gas anesthesia (Isoflurane 1.5–2.5%; Pfizer) and injected with an antibiotic (Baytrile, 0.5 ml, 2%; Bayer Yakuhin), a steroidal anti-inflammatory drug (Dexamethasone; Kyoritsu Seiyaku), an anti-edema agent (Glyceol, 100 μl, Chugai Pharmaceutical) to reduce swelling of the brain, and a painkiller (Lepetan, Otsuka Pharmaceutical). The scalp and periosteum were retracted, exposing the skull, then a 4 mm diameter trephination was made with a micro drill (Meisinger LLC). A 4 mm coverslip (120–170 μm thickness) was positioned in the center of the craniotomy in direct contact with the brain, topped by a 6 mm diameter coverslip with the same thickness. When needed, Gelfoam (Pfizer) was applied around the 4 mm coverslip to stop any bleeding. The 6 mm coverslip was fixed to the bone with cyanoacrylic glue (Aron Alpha, Toagosei). A round metal chamber (6.1 mm diameter) combined with a head-post was centered on the craniotomy and cemented to the bone with dental adhesive (Super-Bond C&B, Sun Medical), mixed to a black dye for improved light absorbance during imaging.

**Viral injections**. A construct used to produce AAV expressing GCaMP8 (pAAV-CAG-GCaMP8) was made based on two plasmids, pAAV-CAG-GFP (#37825, Addgene, Cambridge, MA, USA, a gift from Edward Boyden, MIT, MA, USA) and pN1-GCaMP8 (a kind gift from Junichi Nakai and Masamichi Ohkura, University of Saitama, Saitama, Japan). Solutions including infectious AAV particles were made and purified using a standard method (Tsuneoka et al., 2015)[23]. For imaging experiments, we injected rAAV2/1-CAG-GCaMP8 solution ($2 \times 10^{12}$ gc/ml, 500 nl) into the right visual cortex (AP, −3.3 mm: LM 2.4 mm from the bregma) at a flow rate of ~50 nl/min using a Nanoject II (Drummond Scientific, Broomall, Pennsylvania, USA). Injection depth was 300–350 μm. After confirmation of fluorescent protein expression, we made a craniotomy (~4 mm diameter) centered on the injection point while keeping the dura intact and implanted a cover-glass window, as described above.

**Automated behavioral setup for voluntary head fixation**. The setup has been developed in collaboration with O' Hara & Co., Ltd. (Tokyo) and it is now commercially available via O' Hara & Co., Ltd. (http://ohara-time.co.jp/). The latching unit for physiology has been produced by Micro Industries Co., Ltd. (Tokyo).

**2AFC orientation discrimination task**. Implanted mice were housed individually in standard cages connected to the setups. Visual stimuli were presented on the center of a LCD monitor (33.6 cm × 59.8 cm, 1920 × 1080 pixels, PROLITE B2776HDS-B1, IIYAMA), placed 25 cm in front of mice. The monitor covered ~40° × 100° of visual space including the whole binocular field. Stimuli were Gabor patches, static sine gratings, ~30° in diameter, 0.08 cpd, with randomized spatial phase, and windowed by a stationary two-dimensional Gaussian envelope, which was generated with custom code using the Psychtoolbox extension for Matlab. The stimulus diameter was defined as the 2σ of the Gaussian envelope. The initial version of the software was based on code from Burgess et al.[6], and subsequently customized for this platform.

Most animals exhibited a bias at the beginning of the training. Hence we used an adaptive corrective procedure in which the probability of the target stimulus being presented clockwise $P(C)$ or counterclockwise $P(CC)$ was, (with $P(CC = 1−P (C))$ was calculated to match the probability of previous CC–C choices: $P(C) = \sum_i^N cc_i/(\sum_i^N cc_i + \sum_i^N c_i)$, with $c_i$ and $cc_i = \{1,0\}$, for choosing or not

choosing the corresponding rotation, and $N$ = all previous trials. $P$(C) was updated every 10 trials.

For imaging of GCaMP8 signals, the movement of the gratings on the screen produces strong visually evoked responses. Hence in training and in imaging experiments, to remove this component we introduced an open-loop (non-interactive) period (1.0 s) after the onset of the stimulus. During this period, a rotation of the wheel did not produce any stimulus movement. With training, animals learned to minimize wheel rotations during this period. To test for habituation under the two-photon microscope, we used two mice in addition to the eight reported in Fig. 3. These mice were trained in an earlier version of the same task, where wheel rotations induced L/R translations of the stimuli. This paradigm required a longer training period ($101 \pm 15$ days, $n = 2$), than with c/cc rotations ($26 \pm 7$ days, $n = 8$ mice).

**Go-No-Go auditory detection task**. For the auditory discrimination task, in addition to the setup for visual discrimination task, a speaker (DX25TG59-04, Tymphany) was placed 10 cm in front of the mouse head. Correct detections were rewarded by 4 μl water. A performance index of stimulus detectability was calculated as $d' = z$(Hits fraction) $- z$(FA fraction), with $z$ the inverse of the cumulative Gaussian function. The threshold of wheel rotation to signal an auditory stimulus detection was 70°. The whole setup was enclosed in a $50 \times 50 \times 45$ cm box with sound isolation panels (Sound Guard W, Yahata-Neji).

**Behavioral data analysis**. We fitted the animal's probability of making a right side choice as a function of task difficulty using a psychometric function $\psi$ (Wichmann and Hill; psignifit version 3 toolbox for MATLAB http://bootstrap-software.com/):

$$\psi(\epsilon; \alpha, \beta, \gamma, \lambda) = \gamma + (1 - \gamma - \lambda)F(\epsilon; \alpha, \beta),$$

where $F(x)$ is a cumulative Gaussian function, $\alpha$ and $\beta$ are the mean and s.d., $\gamma$ and $\lambda$ are left and right (L/R) lapsing rates, $\epsilon$ is the signed trial easiness. Confidence intervals were computed via parametric bootstrapping (999 bootstraps). Time-out trials were excluded from the analysis.

We quantified the animals' performance during learning (Fig. 3) by analyzing how the slope, lapse-rates, and bias of the psychometric function $\psi$ changed with training[24].

$$\text{Slope} = \frac{1}{\beta\sqrt{2\pi}}$$
$$\text{Lapsing rate} = \psi(-45) - \psi(45) + 1$$
$$\text{Bias} = \psi^{-1}(0.5)$$

Goodness-of-the-fit was tested separately for each curve by computing the deviance ($D$) and correlation ($r$) within the 95% confidence interval[14]. With only two conditions, the psychometric model is under-constrained, however a (constrained) linear model produced similar results for bias and lapsing rate (defined as in the psychometric model), and for the slope parameter derived from the slope of the fitted line.

**Two-photon imaging**. Imaging was performed using the two-photon imaging mode of the multiphoton confocal microscope (Model A1RMP, Nikon, Japan). The microscope was controlled by A1 software (Nikon). The objective was a ×10 air immersion lens (NA, 0.45; working distance, 4 mm; Nikon). The field of view ($512 \times 512$ pixels) was 850 μm × 850 μm. GCaMP8 was excited at 920 nm and laser power was 10–25 mW. Images were acquired continuously at ~15 Hz frame rate using a resonant scanner. In every imaging session, a vascular image was captured at the surface of the cortex as a reference for imaging field location.

**Analysis of two-photon data**. All the analyses except for neuronal segmentation were done using custom code written in Matlab. Spatial shifts ($x$–$y$ translation) due to movements of the mouse were initially corrected using the $x$–$y$ coordinates of the peak of the spatial cross correlation between a reference frame (average of initial 10 frames) and all the other frames (typically $1$–$2 \times 10^4$ frames). A semi-automatic segmentation of regions of interest (ROIs) was then performed in the motion-corrected data using the Suite2P toolbox (https://github.com/cortex-lab/Suite2P). A neuropil region was also determined automatically for each ROI as a surrounding region (2–8 μm from each ROI) that does not include the soma of other ROIs. The averaged fluorescence signal of each ROI was corrected by the averaged signal of the corresponding neuropil region as $F_{soma} - 0.7 \times (F_{neuropil} - \text{median}(F_{neuropil}))$[25]. dF/F0 was calculated for each ROI. F0 was the average of 10 frames of the neuropil-corrected signal immediately before the onset of visual stimulation (baseline fluorescence). For orientation tuning curves (Fig. 5e), dF/F0 of 0–1 s from the onset of visual stimulus was averaged for each stimulus orientation.

**Data availability**. The technical-drawing data are available in DXF format at https://datadryad.org/ doi: doi:10.5061/dryad.1qv5t. All relevant data are available from the corresponding author upon request.

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

## Acknowledgements

We thank Edward Boyden for donating the plasmids pAAV-CAG-GFP (#37825 Addgene). We thank Junichi Nakai and Masamichi Ohkura at Saitama University for donating pN1-GCaMP8 plasmids. We thank O'Hara & Co. Ltd. for several hardware and software implementations of the main setup. Original technical drawings in Figs. 1, 2, 5a, and Supplementary Figs. 1–6 have been edited with permission from O' Hara & Co., Ltd. We thank Micro Industries Co., Ltd for help with 3D figures. We thank Rie Nishiyama and Yuka Iwamoto for their technical support with the animal facility, surgeries, and viral injections. We thank Matteo Carandini, Kenneth Harris, and Chris Burgess for

sharing hardware information and an initial version of the software for behavioral training. We thank Charles Yokoyama for valuable feedback and editing of the manuscript. This work was supported by RIKEN-BSI institutional funding; JSPS Grants 26290011, 16K18372, and 15H06861; RIKEN Special Postdoctoral Researchers Program.

## Author contributions

R.A. was involved in all stages of the setup's development, designed the auditory task, planned, and supervised the mouse training. T.T. designed the visual task, planned and supervised the animal training, and performed the two-photon experiments. Y.G. was involved in all parts of the setup's development, coordinated the work between the lab and O'Hara & Co. Ltd., wrote the code for the visual and auditory tasks, and helped designing the controller's software. A.B. came up with the idea, designed the setup, supervised all aspects of the project, and wrote the manuscript.

## Additional information

**Competing interests:** The authors declare competing financial interests: RIKEN-sponsored patent JP2016-129406 (Japan validity only).

