## [Peer Review File · Nature Communications]

Editorial Note: This manuscript has been previously reviewed at another journal that is not operating a transparent peer review scheme. This document only contains reviewer comments and rebuttal letters for versions considered at Nature Communications. Parts of this peer review file have been redacted as indicated to maintain the confidentiality of unpublished data.

Reviewers' comments:

Reviewer #3 (Remarks to the Author):

By reducing the scope of their study to a technical report, Aoki et al. have satisfied my major concerns. I only have a few minor concerns remaining:

1. Line 10 of the abstract should read: "we coupled the platform to a two-photon microscope"
2. A large part of the justification in the introduction for using this automated design is to facilitate comparisons across labs through standardization. While this is a worthy goal, I do not think this either a likely outcome of this paper nor the major value of this approach. For one, all labs would have to adopt this particular interface; and even if they did, the setup still leaves room for task-design and training variability that will make direct comparisons across labs difficult. The major value is the minimal experimenter intervention that therefore increases training output and standardizes procedures across conditions within the lab.
3. Figure 4c suggests that naive mice have a higher FA rate than Hit rate on the auditory detection task. This is surprising- is there a feature of the task/training that makes FAs more likely than Hits? Figure 4c also suggests that mice have a higher FA rate after they have learned the task than before. Thus, it is not clear that the mice have actually learned the nogo portion of the go-nogo task.
4. The final paragraph of the discussion makes the argument that self-initiation of trials is actually unwanted since animals could earn all of their water too quickly. However, this assumes an unnecessary tradeoff, since the experimenter could still control reward volume and trial availability in a self-initiated task. Self-initiation is useful to discriminate misses and CRs in a go-nogo task from satiation.

Reviewer #4 (Remarks to the Author):

In their work titled "An Automated High-Throughput Platform for Mouse Behavior and Physiology with Voluntary Head-fixation" the authors present an automated training system for mice which incorporates voluntary head fixation with the capability of being a component in a high throughput training setup. They demonstrate the ability to integrate the system in different task structures, e.g. 2AFC and Go/No-go, and different sensory modalities, e.g. visual and auditory. They also state that their setup would be compatible with electrophysiology recording systems requiring head fixation (though they offer no demonstration of this, I see no reason this would not be possible), optogenetic techniques (I'm assuming they're referring to single cell manipulations or targeted approaches where the experimenter has access to the entire dorsal cortical surface through a cranial window, but references are lacking in this regard), and 2-photon calcium imaging (which they do demonstrate here). While there is much to like in this work, automated high-throughput voluntary head-fixation is an exciting and useful technology, it also attempts to solve a problem of lack of reproducibility and standardization across labs by adding yet another hardware and software system into an already crowded pool. It's not just a matter of demonstrating that a new system is superior to existing setups (more on that below) but it must be vastly superior to overcome the inertia and motivate a lab to switch their existing setups to this technology (a point which I feel the authors ignore). Beyond that I have a number of additional concerns which I feel must be addressed before I can recommend publication.

1) Many places throughout the manuscript there appears some awkward grammar, e.g. line 1: "Recording neural activity during model animal behavior...", line 10: "...we couples the platform to two-photon microscope...", and line 12: "...will be an effective prototype for next generation of mouse cognitive studies.". I will not outline them all here, and will leave it to the editor to work with the

authors to improve this issue.

2) As stated above the authors are attempting to solve the problem of a diverse array of hardware and software across labs which admittedly does lead to data sharing and potentially reproducibility issues, by themselves developing yet another hardware and software system. They state in the abstract that there is “an absence of tools that can generate large, sharable datasets for the research community in a time and cost effective way.” Yet the authors themselves are aware of and cite the Bcontrol system developed more than a decade ago at Cold Spring Harbor Laboratory. This system is currently in use in many labs across the world generating many hundreds of thousands of behavioral trials daily, and is integrated with ephys, optogenetic, and 2-photon experiments. Operating in Matlab BControl data is therefore presumably as sharable as the data presented here. A newer generation of automated training and behavior control from the Kepecs Lab, BPod, is also gaining wide usage offering a system with low cost open source technology. Even voluntary head fixation, at least in rats, is an established technology (Scott, Brody, and Tank, Neuron 2013). That system has two substantial benefits to this system: 1) the head plate is automatically locked in place with pneumatic pistons as opposed to here where an experimenter must tighten a set of four screws for imaging sessions (i.e. not a fully automated system); and 2) their kinematic mount allows for micron precision registration whereas here there is no statement as to registration reliability. What I see as the only true advance here beyond it being voluntary head fixation in a novel species, is the automated way in which this setup is tied into the animal’s home cage allowing for head fixation performance without an experimenter ever having to handle the animal (which admittedly can have enormous benefits for mouse behavioral performance, but none is quantitatively demonstrated here).

3) While the authors state this work was approved by their local IACUC I have a number of concerns which I know my local IACUC would raise. First, when introducing the head-restraining device into their home cage, how frequently is the animal checked on and is there any way their head plate can become stuck in any crevice in the device. For example, we’re banned from having food hoppers in cages with animals with implants due to the possibility that their implant can become stuck in the gratings in the hopper wall. Since training sessions are short the head fixation is less of a concern there since presumably a human will observe the animals between sessions. There too however, I am concerned regarding the automated doors that connect the home cage to the training setup. While the authors state these doors are under the control of sensors that determine the position of the animal in the setup, such technology can fail. Do these doors have a safety feature that can measure resistance and reopen if they detect an object stuck in them?

4) While the authors state on line 196 “Using vascularization landmarks we could image the same cells over days or weeks...” there is some important information that is lacking which would allow readers to accurately assess this technology. 1) What is the registration error between sessions? 2) How is the matching z depth found between sessions? 3) How are movement artifacts corrected for during a session? 4) Quantification of the positional stability of the setup during an imaging session. 5) What criteria are used to identify the same cell across sessions?

5) On line 241 the authors state that self-initiation of trials and self-detaching of head fixation are undesirable since they could lead to unmotivated subjects performing at chance levels. While I understand the need for experimenter control of trial timing and fixation disengagement in certain circumstances this is in no way a universal truth. There are hundreds of animal studies where trials are initiated by the subject which demonstrate high performance levels across many hundreds of trials on a range of cognitively demanding tasks. While self-initiated disengagement from head fixation is a newer technology it too is not an impediment to good performance. These are limitations of the current system presented by the authors and should be presented as such, and not justified as requirements for good behavioral performance since such a statement runs counter to decades of

literature.

6) Figure 5D. With such large scale imaging of V1 why show cell response as a function of the animal's choice rather than what should be more clearly encoded, orientation tuning.

7) While I appreciate the degree of detail offered in the supplementary material regarding the schematics of the system I have two large concerns. 1) Is this setup fully open source? While I appreciate that some labs may want to purchase a pre-built system, many others would want to construct it themselves given suitable blueprints and material lists. 2) The statement on line 276 that the system is commercially available necessitates a conflict of interest statement.

Minor Issues:

1) On line 104 the authors appear to understate the capability of their system "Hence a single setup can train 4 mice in 24 hours." If each mouse is trained for 20-30 minutes twice a day then 4 mice at most require 4 hours of time in the rig (unless I am misunderstanding something). Why waste the other 20 hours? However on line 235 they state "...we could produce an average of 1 trained mouse every 1.3 hours (i.e. 2 days)." This is misleading. If I understand correctly the statement "a training rig can produce an average of 1 trained mouse every 2 days (i.e. 1.3 hours of rig time)" would be more accurate.

2) The statement on line 132-134 appears counter-productive to the authors' stated goal of generating large datasets. I understand the need to limit an animal's water intake to keep them motivating and performing at a high degree, but why not simply reduce the reward drop size and thereby allow them to perform more correct trials in a session.

3) There is disagreement throughout the paper as to how many sessions per day the animal performs, i.e. 2 or 3.

4) The authors should specify in the text on line 142 which psychometric parameters they're referring to.

5) Does this setup allow for group housed animals or must they be singly housed? Housing status should be clearly indicated in the Methods.

6) For the auditory discrimination task it's unclear how the sound attenuation chamber is coupled to the mouse's home cage. Is there simply a hole in the side of the sound chamber through which the mouse tunnel is connected? Does this affect the sound attenuation level?

7) Line 319 it appears the authors had a placeholder for a reference "(cite)" which they did not fill.

8) Figure 1 and supplementary figures including schematics. I presume the numbers are dimensions, possibly millimeters, but that should be made clear in the figure legends. Line 396 references "...dotted lines on the side of the tube...", however I'm unable to see what they're referring to in the figure.

9) Figure 5C the example cells do not print clearly and are barely identifiable on a screen.

Reviewer #3 (Remarks to the Author):

By reducing the scope of their study to a technical report, Aoki et al. have satisfied my major concerns. I only have a few minor concerns remaining:

1. Line 10 of the abstract should read: "we coupled the platform to a two photon microscope"

We have fixed it

2. A large part of the justification in the introduction for using this automated design is to facilitate comparisons across labs through standardization. While this is a worthy goal, I do not think this either a likely outcome of this paper nor the major value of this approach. For one, all labs would have to adopt this particular interface; And even if they did, the setup still leaves room for task design and training variability that will make direct comparisons across labs difficult. The major value is the minimal experimenter intervention that therefore increases training output and standardizes procedures across conditions within the lab.

We mostly agree with the reviewer's opinion. However, we continue to consider the large-scale adoption of this setup a feasible plan, together with the standardization of a few task designs for open access and freely sharable data. [Redacted] Nevertheless, we agree with the reviewer that a parsimonious, and realistic justification of our goal is preferable, so we have amended the text in the Introduction to better emphasize the benefits of our platform for within-lab standardization:

Ln17: "the specificities of these paradigms and their integration with the growing array of state-of-the-art brain physiological recording systems differ greatly among and within laboratories due to the variability introduced by the experimenter's intervention."

Ln34: "From a research economics perspective, an ideal mouse system would feature self head-fixation for behavioral training and rapid exploration of a large space of complex behavioral parameters with minimal experimenter intervention, allow high-throughput automated training, have the capability to explore various sources of psychometric data, flexibly integrate multiple physiology recording/stimulation systems, and enable the efficient generation of large, sharable, and reproducible datasets to standardize procedures within, and across laboratories."

Ln 213: "[...]with the drawback of hindering the within- and across-labs reproducibility"

3. Figure 4c suggests that naïve mice have a higher FA rate than Hit rate on the auditory detection task. This is surprising, is there a feature of the task/training that makes FAs more likely than Hits? Figure 4c also suggests that mice have a higher FA rate after they have learned the task than before. Thus, it is not clear that the mice have actually learned the nogo portion of the go-nogo task.

We thank the reviewer for this insightful observation. We re-analyzed the behavioral data with the inclusion of several new sessions and found that the already small difference in FA rates from naïve to expert animals was not statistically significant (new Figure 4C). Although the FA rates did not change, it remains true that FA rates in naïve mice are higher than Hit rates (statistically significant, $p < 0.005$, Wilcoxon rank sum test). There is nothing in the task structure that could cause this, but one possible explanation might be related to the 'go' stimulus itself. Direct observations of the animals' behavior suggest that naïve mice are at times startled by the go-sound, leading to reduced wheel movements during the sound presentation. This observation has been made before in mice with similar sound pressure level and frequency (80 dB, 8 kHz) and described as an "acoustic startle response" (or reflex) - ASR, characterized by rapid muscle contraction (Szabo I, 1964) and arrest of ongoing movements (Graham FK, 1979), and reported more recently in Curzon P, et al., (2009), Popelář J, et al., *Neurosci. Letters* (2013). We included this observation in the Results, Ln 168: "FA rates remained constant throughout training, with FA rates in naïve mice higher than Hit rates ($p < 0.005$, Wilcoxon signed rank test) possibly due to a startle reflex following the target sound presentation¹⁸⁻²¹".

4. The final paragraph of the discussion makes the argument that self-initiation of trials is actually unwanted since animals could earn all of their water too quickly. However, this assumes an unnecessary tradeoff, since the experimenter could still control reward volume and trial availability in a self-initiated task. Self-initiation is useful to discriminate misses and CRs in a go-nogo task from satiation.

We agree with this criticism and accordingly we have significantly reduced the scope of our observation, and we suggest the possibility of recording/imaging paradigms requiring trial durations set by the experimenter. The text now reads as follows: “[...] (2) control of the frequency and duration of the trials in those behavioral and physiology paradigms demanding session’s durations set by the experimenter rather than by the animal” We have deleted the rest.

Reviewer #4 (Remarks to the Author):

In their work titled “An Automated High Throughput Platform for Mouse Behavior and Physiology with Voluntary Head fixation” the authors present an automated training system for mice which incorporates voluntary head fixation with the capability of being a component in a high throughput training setup. They demonstrate the ability to integrate the system in different task structures, e.g. 2AFC and Go/Nogo, and different sensory modalities, e.g. visual and auditory. They also state that their setup would be compatible with electrophysiology recording systems requiring head fixation (though they offer no demonstration of this, I see no reason this would not be possible), optogenetic techniques (I’m assuming they’re referring to single cell manipulations or targeted approaches where the experimenter has access to the entire dorsal cortical surface through a cranial window, but references are lacking in this regard)

We have added the relevant references (Ln 26):

Rickgauer, J.P., Deisseroth, K. & Tank, D.W. Simultaneous cellular-resolution optical perturbation and imaging of place cell firing fields. *Nat Neurosci* 17, 1816-24 (2014).

Packer, A.M., Russell, L.E., Dagleish, H.W. & Hausser, M. Simultaneous all-optical manipulation and recording of neural circuit activity with cellular resolution in vivo. *Nat Methods* 12, 140-6 (2015).

Then we cited them again in the Discussion, Ln 220 “Head fixation is also desirable for optical/optogenetic technologies aiming to achieve cellular-level resolution^{1,2}”

[...] and 2photon calcium imaging (which they do demonstrate here). While there is much to like in this work, automated high throughput voluntary head fixation is an exciting and useful technology, it also attempts to solve a problem of lack of reproducibility and standardization across labs by adding yet another hardware and software system into an already crowded pool. It’s not just a matter of demonstrating that a new system is superior to existing setups (more on that below) but it must be vastly superior to overcome the inertia and motivate a lab to switch their existing setups to this technology (a point which I feel the authors ignore).

We agree that overcoming inertia and convincing laboratories to adopt our setup is not a trivial goal. Here we’d like to emphasize a couple of points: although it is a fact that several labs are interested in automated voluntary head-fixation, there are currently very few labs already using such technology (only one for mice to our knowledge, but with no demonstration of behavioral training capability), so in our view the key issue is not so much convincing several labs to ‘switch’ their setups, but rather to invite the much larger majority of labs currently without a setup to adopt ours. We would also like to emphasize that we are not trying to convince readers that our setup is better or preferable to Prof. Murphy’s. As explained below, and in the manuscript, differences among our setups are such that the users’ choice will depend on their specific research needs. [Redacted]

[Redacted].

Beyond that I have a number of additional concerns which I feel must be addressed before I can recommend publication.

1) Many places throughout the manuscript there appears some awkward grammar, e.g. line 1: "Recording neural activity during model animal behavior...", line 10: "...we couples the platform to two photon microscope...", and line 12: "...will be an effective prototype for next generation of mouse cognitive studies.". I will not outline them all here, and will leave it to the editor to work with the authors to improve this issue.

We apologize for the awkward grammar and we have asked English native speakers to proof-read the manuscript

2) As stated above the authors are attempting to solve the problem of a diverse array of hardware and software across labs which admittedly does lead to data sharing and potentially reproducibility issues, by themselves developing yet another hardware and software system. They state in the abstract that there is "an absence of tools that can generate large, sharable datasets for the research community in a time and cost effective way." Yet the authors themselves are aware of and cite the Bcontrol system developed more than a decade ago at Cold Spring Harbor Laboratory. This system is currently in use in many labs across the world generating many hundreds of thousands of behavioral trials daily, and is integrated with ephys, optogenetic, and 2photon experiments. Operating in Matlab BControl data is therefore presumably as sharable as the data presented here. A newer generation of automated training and behavior control from the Kepecs Lab, BPod, is also gaining wide usage offering a system with low cost open source technology. Even voluntary head fixation, at least in rats, is an established technology (Scott, Brody, and Tank, Neuron 2013). That system has two substantial benefits to this system: 1) the head plate is automatically locked in place with pneumatic pistons as opposed to here where an experimenter must tighten a set of four screws for imaging sessions (i.e. not a fully automated system); and 2) their kinematic mount allows for micron precision registration whereas here there is no statement as to registration reliability. What I see as the only true advance here beyond it being voluntary head fixation in a novel species, is the automated way in which this setup is tied into the animal's home cage allowing for head fixation performance without an experimenter ever having to handle the animal (which admittedly can have enormous benefits for mouse behavioral performance, but none is quantitatively demonstrated here).

We largely agree with the reviewer's comments and realized that we did not sufficiently explain some of the key innovative elements of this setup. In agreement with the reviewer, an important innovation is the voluntary head fixation in a novel species. However, we'd like to emphasize that this is not merely a novel species, but also by our intention the mammalian species with the largest genetic toolbox (Ln 23). Another point acknowledged by the reviewer is the absence of the experimenter's intervention. Importantly, we are not claiming that automated voluntary head-fixation is categorically superior to freely-moving solutions (e.g. BControl and BPod): the two implementations address very different technical challenges and serve different research purposes. Here, we are only proposing a setup for labs specifically interested in a platform with automated voluntary head-fixation in mice, and we believe there are quite a few interested in this technology. This scope issue is discussed in Ln 218: "[...] stable head fixation, an important feature for behavioral assays relying on accurate measurement of many sensory modalities, such as vision that requires eye tracking and view-point stability. Head fixation is also desirable for optical/optogenetic technologies aiming to achieve cellular-level resolution^{1,2}". Another important point is the flexible use of this head-fixed setup with multiple physiology recording systems. Regarding the platforms with voluntary head fixation, the excellent system by Scott, Brody, and Tank features automatic 2P imaging, while the system by Murphy et al., integrates automatic widefield imaging (with no demonstration of behavioral training). Our platform sacrifices imaging automation in favor of flexible integration with 2P, widefield, optogenetic devices, and electrophysiology systems (not demonstrated in the manuscript); these are the technologies currently in use in our lab to record from trained mice for different levels of analysis and types of research projects. We are not claiming that this

solution is better or more favorable than others. However, the capacity for flexible integration with automated voluntary head-fixation is objectively a novel element that we believe researchers might find interesting or preferable to other solutions. We have made this point clearer by amending the text in several sections. In the introduction, Ln 34:

“From a research economics perspective, an ideal mouse system would feature self head-fixation for behavioral training and rapid exploration of a large space of complex behavioral parameters with minimal experimenter intervention, allow high-throughput automated training, have the capability to explore various sources of psychometric data, flexibly integrate multiple physiology recording/stimulation systems [...]”

The same point is then reiterated in the section on the Latching Unit for Physiology, Ln 173: “In order to serve as a powerful tool to study the neural basis of cognitive functions, a setup must not only enable behavioral training but also (1) be easily integrated with diverse customized physiology setups,...”, and in a few more parts in the Discussion (e.g. Ln: 212, Ln 223).

We have also toned down the initial statement in the Discussion, which now reads as follows: “The traditional experimental approach to integration of an animals’ behavioral training and physiological recording has often resorted to lab-specific experimental configurations relying on the experimenter’s intervention with the drawback of hindering within- and across-labs reproducibility.”, thus rectifying the previous reference to “traditional approaches to brain research”.

Following also a suggestion by the third reviewer we have amended the introduction by emphasizing the advantages of within-lab standardization, and thus deemphasizing the statement cited by the reviewer on the “*absence of tools that can generate large, sharable datasets for the research community in a time and cost effective way*”, and the “*problem of a diverse array of hardware and software across labs*” Ln17: “the specificities of these paradigms and their integration with the growing array of state-of-the-art brain physiological recording systems differ greatly among and within laboratories due to the variability introduced by the experimenter’s intervention.”

Ln39: “[...] the efficient generation of large, sharable, and reproducible datasets to standardize procedures within, and across laboratories.”

Similarly, Ln 213: “with the drawback of hindering the within- and across- labs reproducibility”

3) While the authors state this work was approved by their local IACUC I have a number of concerns which I know my local IACUC would raise. First, when introducing the head restraining device into their home cage, how frequently is the animal checked on and is there any way their head plate can become stuck in any crevice in the device. For example, we’re banned from having food hoppers in cages with animals with implants due to the possibility that their implant can become stuck in the gratings in the hopper wall. Since training sessions are short the head fixation is less of a concern there since presumably a human will observe the animals between sessions. There too however, I am concerned regarding the automated doors that connect the home cage to the training setup. While the authors state these doors are under the control of sensors that determine the position of the animal in the setup, such technology can fail. Do these doors have a safety feature that can measure resistance and reopen if they detect an object stuck in them?

These are certainly important concerns. In Japan we are regulated by the 1973 animal welfare act, amended several times in the last few decades and now closely matching the US or UK standards (the senior author is familiar with these regulations having been licensed for animal experiments in all of these countries). Accordingly, to be approved in Japan we had to demonstrate our setup would not cause any harm to the animals and actually facilitate the implementation of the three Rs of animal testing. Thanks to sensor technology (with no “resistance detectors” on the doors), the careful design of the head chamber (Supp. Fig. 1) and software (see the “emergency” calls described in the pseudo-code, Supp. Table 3) we haven’t had animal injuries across ~100 mice that have gone through our setups. This

is one statistic we can objectively report, but we clearly cannot exclude with absolute confidence that mechanical failures one day could occur and possibly cause an unforeseen accident. We now refer to these observations in the Results, Ln 111: “The hardware and software design has proved to be safe for the animals with no reported accidents over ~100 mice trained in the setups to date”.

4) While the authors state on line 196 “Using vascularization landmarks we could image the same cells over days or weeks...” there is some important information that is lacking which would allow readers to accurately assess this technology. 1) What is the registration error between sessions? 2) How is the matching z depth found between sessions? 3) How are movement artifacts corrected for during a session? 4) Quantification of the positional stability of the setup during an imaging session. 5) What criteria are used to identify the same cell across sessions?

Indeed, several details related to imaging stability and the registration analysis were missing. We have amended the text (in the Methods, “Analysis of two-photon data”) and added an explanatory figure (Supplementary Fig7, and further explanations in its caption). Specifically, for the registration, we now explain how after an initial estimation of matching regions by the experimenter using vascularization landmarks, we then resort to a method that uses spatial correlations as a metric to accurately register (off-line) the images. As explained in Supplementary Figure 7A-E, for the across-sessions image registration we compared two methods: one based on spatial correlations and the other on a more sophisticated optimizer using Mattes mutual information, and correcting for scaling, rotation, and possibly shear. The two methods produced similar results and both indicate registration errors typically smaller than half the size of a cell’s soma (Supplementary Fig7D). Identification of same cells across sessions was done by off-line analysis (Supplementary Fig7E). We applied a semi-automated segmentation algorithm independently to each imaging sessions. Then ROIs were matched across different sessions based on a criterion for ROIs overlap: overlapping area $> 0.5 \times$ area in the former session. This identification procedure is explained in the figure’s caption. For the within-session corrections we used the faster spatial-correlation method to quantify the frame-by-frame corrections, typically $\sim 2\mu\text{m}$ (Supplementary Fig7F). Using information about the surface area of the segmented somas, we also argue that we can accurately match the z-depth as well. Depth estimation is primarily based on visual inspection during the experiment by comparing the online image from the microscope to a reference image acquired days before. Visual inspection is faster than an automated approach with online quantifications of the registration errors, and speed is very important during the experiment. Indeed, several recording protocols are in place during an imaging session, and it is advisable to keep the animal head-fixed ideally for less than ~ 1 hour. Hence the experimenter has to quickly determine the ROI and depth of focus via visual inspection. The quantification of this procedure can then be evaluated off-line after the imaging session. This is now explained in the captions of Supplementary Fig 7.

5) On line 241 the authors state that self-initiation of trials and self-detaching of head fixation are undesirable since they could lead to unmotivated subjects performing at chance levels. While I understand the need for experimenter control of trial timing and fixation disengagement in certain circumstances this is in no way a universal truth. There are hundreds of animal studies where trials are initiated by the subject which demonstrate high performance levels across many hundreds of trials on a range of cognitively demanding tasks. While self-initiated disengagement from head fixation is a newer technology it too is not an impediment to good performance. These are limitations of the current system presented by the authors and should be presented as such, and not justified as requirements for good behavioral performance since such a statement runs counter to decades of literature.

We agree with this observation and corrected the scope of our observation; we also note the possibility of recording/imaging paradigms requiring trial durations set by the experimenter. The text now reads as

follows: “[...] (2) control of the frequency and duration of the trials in those behavioral and physiology paradigms demanding session durations set by the experimenter rather than by the animal” We have deleted the rest.

6) *Figure 5D. With such large scale imaging of V1 why show cell response as a function of the animal's choice rather than what should be more clearly encoded, orientation tuning.*

We thought that readers might be interested in response properties linked specifically to the behavioral task across days. However we agree with the reviewer that the stability of the orientation tuning could be an excellent proxy for the functional stability of our preparation. We have amended the figure accordingly and we now show both the responses to L/R choices and orientation tuning (new Figure 5E). We hope the reviewer will find this compromise acceptable. As a side observation, we would like to note that when using viral methods the expression level can vary significantly across weeks, so if a study aims specifically to address the tuning stability across days, transgenic animals would certainly be preferable to viral tools. Here we wanted to demonstrate that the setup has imaging resolution and stability that allows recording from the same cells over days. However, we did not want to make a strong statement specifically on the functional stability of such responses. We have made this point more explicit by amending the text on Ln 204: “As a corollary of this cellular-level stability resolution, our semi-automated procedure ...”

7) *While I appreciate the degree of detail offered in the supplementary material regarding the schematics of the system I have two large concerns. 1) Is this setup fully open source? While I appreciate that some labs may want to purchase a prebuilt system, many others would want to construct it themselves given suitable blueprints and material lists. 2) The statement on line 276 that the system is commercially available necessitates a conflict of interest statement.*

We agree this is an important point. It's in our best interest to make this platform useful and to provide all necessary information to users for its implementation. Pending RIKEN's approval, we are considering to deposit software and technical drawings in an open-access website. The company that helped us develop the product is certainly willing to provide support if/when contacted. We have added a conflict of interest statement as well.

Minor Issues:

1) *On line 104 the authors appear to understate the capability of their system “Hence a single setup can train 4 mice in 24 hours.” If each mouse is trained for 20-30 minutes twice a day then 4 mice at most require 4 hours of time in the rig (unless I am misunderstanding something). Why waste the other 20 hours? However on line 235 they state “...we could produce an average of 1 trained mouse every 1.3 hours (i.e. 2 days).” This is misleading. If I understand correctly the statement “a training rig can produce an average of 1 trained mouse every 2 days (i.e. 1.3 hours of rig time)” would be more accurate.*

The reviewer is correct and we have changed the text as indicated (Ln 241). Furthermore, we also refer to the possibility of having less idle time, as suggested by the reviewer as well (Ln 110): “At a current capacity of 12 platforms, we can train 48 mice/day and collect up to ~12000 trials/day. This number could easily be increased if the cages were replaced more frequently.”

2) *The statement on line 132-134 appears counterproductive to the authors' stated goal of generating large datasets. I understand the need to limit an animal's water intake to keep them motivating and performing at a high degree, but why not simply reduce the reward drop size and thereby allow them to perform more correct trials in a session.*

Also here we agree with the reviewer's observation: reducing the reward drop size is indeed a possibility. We have deleted the statement.

3) *There is disagreement throughout the paper as to how many sessions per day the animal performs, i.e. 2 or 3.*

We have corrected this discrepancy.

4) The authors should specify in the text on line 142 which psychometric parameters they're referring to.

We fixed it.

5) *Does this setup allow for group housed animals or must they be singly housed? Housing status should be clearly indicated in the Methods.*

We did specify this point on Ln 50, but we are now reporting this information in the methods as well, Ln 284.

6) *For the auditory discrimination task it's unclear how the sound attenuation chamber is coupled to the mouse's home cage. Is there simply a hole in the side of the sound chamber through which the mouse tunnel is connected? Does this affect the sound attenuation level?*

We have added the following text (Ln 153): "The transparent PVC tube connecting the cage to the main setup passed through an aperture on the side of the isolation box". This certainly affected the attenuation level but it seemed good enough for this task.

7) *Line 319 it appears the authors had a placeholder for a reference "(cite)" which they did not fill.*

We fixed it.

8) *Figure 1 and supplementary figures including schematics. I presume the numbers are dimensions, possibly millimeters, but that should be made clear in the figure legends. Line 396 references "...dotted lines on the side of the tube...", however I'm unable to see what they're referring to in the figure.*

We have added this information, now at the end of the caption of Figure 1 and in all captions in the supplementary Figures. We have removed the statement on the dotted lines. It was indeed a mistake related to a previous version of the Figure.

9) *Figure 5C the example cells do not print clearly and are barely identifiable on a screen.*

We have fixed this problem in the new version of the Figure.

REVIEWERS' COMMENTS:

Reviewer #4 (Remarks to the Author):

I would like to thank the authors for extensively and satisfactorily addressing all of my concerns. Upon rereading the revised manuscript along with the supplement I have no further concerns and can now recommend the manuscript be published with no further revisions.